# Online Convex Optimization with Hard Constraints: Towards the Best of Two Worlds and Beyond

**Hengquan Guo**
ShanghaiTech University
guohq@shanghaitech.edu.cn

**Xin Liu** *
ShanghaiTech University
liuxin7@shanghaitech.edu.cn

**Honghao Wei**
University of Michigan, Ann Arbor
honghaow@umich.edu

**Lei Ying**
University of Michigan, Ann Arbor
leiying@umich.edu

## Abstract

This paper considers online convex optimization with hard constraints and analyzes achievable regret and cumulative hard constraint violation (violation for short). The problem distinguishes itself from online convex optimization with soft constraints, where a violation at one round can be compensated/cancelled by a conservative decision at a different round. We propose a RECtified Online Optimization algorithm (RECOO) and consider two settings: fixed constraints and adversarial constraints. Both settings have been considered in the literature. Compared with existing results, *RECOO achieves the best of two worlds and beyond.* For the fixed-constraints setting, RECOO achieves $O\left(\sqrt{T}\right)$ regret and $O(1)$ violation, where $T$ is the learning horizon. The best known results in this case are $O(\sqrt{T})$ regret and $O\left(T^{1/4}\right)$ violation. For the adversarial-constraints setting, it guarantees $O(\sqrt{T})$ regret and $O(T^{3/4})$ violation, which match the best existing results. When the loss function is strongly convex, RECOO can guarantee $O(\log T)$ regret and $O(1)$ violation for fixed constraints, and $O(\log T)$ regret and $O(\sqrt{T \log T})$ violation for adversarial constraints. Both these results are order-wise better than the existing bounds. The regret and violation bounds mentioned above use the best fixed decision in hindsight as the baseline. This paper further considers a dynamic baseline where the comparator sequence is time-varying. This paper shows that RECOO not only improves the existing bounds for the fixed-constraints setting but also *for the first time,* establishes dynamic regret and violation bounds for the adversarial-constraints setting. Our experiment results confirm that RECOO outperforms several existing algorithms for both fixed and adversarial constraints.

## 1 Introduction

Online convex optimization (OCO) is a general framework for modelling and studying online decision-making in uncertain environments [24, 10, 21], where the learner adapts its decisions to minimize a loss function or to maximize a utility function when interacting with the environment in real-time. OCO has a broad range of applications such as online advertising [18, 3], resource allocation in network systems [9, 2, 33], load balancing in queueing systems [13, 15], and personalized healthcare [36, 26], etc. Specifically, given an unknown convex loss function $f_t(\cdot)$, the learner makes decision $x_t$ at each round to minimize the total loss over $T$ rounds, i.e. $\min_{x \in \mathcal{X}} \sum_{t=1}^{T} f_t(x_t)$, where the loss

---

*Corresponding author

36th Conference on Neural Information Processing Systems (NeurIPS 2022).

of decision $x_t$ is revealed after $x_t$ is executed. In practice, the decisions are often subject to a variety of operational constraints. For example, in online advertisement systems, users, who submit bids to advertise their items in order to maximize their click-through-rates, often have a weekly or monthly budget; and in a patient onboarding system, the hospital assigns an incoming patient to a medical unit to optimize the quality of treatment subject to personnel and medical resource constraints. To consider these applications, we study constrained online convex optimization (COCO) with $g_t(\cdot)$ being a convex constraint function and $g_t(x_t) \leqslant 0$ being the constraint at round $t$. In COCO, the learner aims at minimizing the total loss while keeping the constraint violation to its minimum.

One possible method to satisfy the constraints is to project the decision $x_t$ into the feasible set at each round [10]. However, such projection may not be possible when the constraints $g_t(\cdot)$ are adversarial and unknown before hand. Even if $g_t(\cdot)$ is known before hand, projection-based methods (e.g., projected online gradient descent) may require heavy computation when $g_t(\cdot)$ is a complicated function because it is equivalent to solving a constrained quadratic optimization problem [12, 17, 34]. This has motivated projection-free online learning methods such as the online Frank-Wolfe algorithm [12] where the projection operator at each round is by a linear programming with the exact same constraint functions. Recently, a sequence of studies have considered COCO with *soft* constraints, where the constraints are allowed to be violated at some rounds as long as they are satisfied in the long term [17, 14, 20, 32, 25, 5, 4, 34, 27] (in other words, the severity of constraint violation is evaluated based on $\sum_t g_t(x_t)$). We call them soft constraints because the constraint violation at one round could be compensated in a different round. In other words, a decision sequence might have zero constraint violation based on the metric despite violating the constraint(s) at almost every round. For example, consider a decision sequence such that $\{g_t(x_t)\} = \{-1000, 1, 1, 1, \ldots, 1, 1\}$ with 1000 ones. For such a sequence, we have $\sum_{t=1}^{\tau} g_t(x_t) \leqslant 0$ for all $\tau \leqslant T$, but the constraint is violated at all rounds except the first round. While some constraints such as budget or fairness constraints are naturally soft constraints, stronger notions of constraint and constraint violation are needed for other applications, in particular, for safety-critical applications. In this paper, we consider a stronger notion of constraint violation as in [35, 30, 31]

$$\mathcal{V}(T) := \sum_{t=1}^{T} g_t^+(x_t), \tag{1}$$

where that the operator $(\cdot)^+ = \max(\cdot, 0)$ is imposed at every round and all violations during the decision process are added up. We call this *hard* constraint and the metric *cumulative hard constraint violation*. If a decision sequence has a small $\mathcal{V}(T)$, it implies the constraints are satisfied most of the time during the learning. For example, consider the same example, $\mathcal{V}(T) = 1000$ instead of 0 under this stronger notion of constraint violation. In this paper, we consider the following two settings:

- Fixed constraints $g_t(x) = g(x), \forall t$, where the constraint function remains the same over time but is not necessarily known to the learner. Note the setting of known and fixed constraints in [14, 17, 30, 34] is a special case of ours.

- Adversarial constraints $g_t(x)$, where the constraint function $g_t(x)$ is unknown when making decision at round $t$ and can be arbitrarily and adversarially chosen, as in [25, 20, 31].

We will show that RECOO is a unifying algorithm that achieves small regret and violation in both settings. Next, we summarize our main contributions and compare them with the most related works.

## 1.1 Main contributions

**Algorithm:** This paper develops a RECtified Online Optimization algorithm, called RECOO, for COCO-Hard (COCO with hard constraints). RECOO is a *unifying* algorithm for both fixed and adversarial constraints. By introducing rectifiers in both "decision-making" and "penalty update" components, RECOO maintains a minimum penalty price for constraint violation and encourages conservative/pessimistic decisions, which is the key to minimizing the cumulative hard constraint violation.

**Regret and violation bounds:** We first consider the best fixed decision in hindsight as the baseline (also called the static baseline). The main theoretical results are summarized below.

- For fixed constraints, RECOO achieves $O(\sqrt{T})$ regret and $O(1)$ violation for convex loss functions, and $O(\log T)$ regret and $O(1)$ violations for strongly-convex loss functions.

- For adversarial constraints, RECOO achieves $O(\sqrt{T})$ regret and $O(T^{3/4})$ violation for convex loss functions, and $O(\log T)$ regret and $O(\sqrt{T \log T})$ violation for strongly convex loss functions.

We then consider a dynamic baseline where the comparator is time-varying with path length $P_T$ and establish the following results.

- For fixed constraints, RECOO achieves $O(\sqrt{P_T T})$ regret and $O(\log T)$ violation, where $P_T$ is the path length of the feasible comparator sequence defined by $P_T = \sum_{t=1}^{T-1} \|y_{t+1} - y_t\|$ with $g_t(y_t) \leqslant 0, \forall t \in [T]$ as in [37, 30].
- For adversarial constraints, RECOO achieves $O\left(P_T \sqrt{T}\right)$ regret and $O(T^{3/4})$ violation.

The comparison with most related works are summarized in Tables 1 and 2, from which we can see that RECOO achieves and improves the state-of-the-art results for both settings ("N/A" means no results to the best of our knowledge). Below are a few highlights we would like to mention:

- For fixed constraints and the static baseline, RECOO not only improves the results in [30] but also answers a conjecture in [30] that $O(1)$ violation may be achievable under Slater's condition (the condition holds when there exist a positive $\epsilon > 0$ and $x \in \mathcal{X}$ such that $g_t(x) \leqslant -\epsilon, \forall t \in [T]$). We proved that the conjecture is true even without Slater's condition.
- For adversarial constraints, the static baseline, and strongly convex loss functions, RECOO reduces the $O(T^c)$ regret and $O(T^{1-c/2})$ violation in [31] to $O(\log T)$ and $O(\sqrt{T \log T})$, respectively.
- For fixed constraints and the dynamic baseline, RECOO improves the violation from $O(\sqrt{T})$ in [30] to $O(\log T)$, while maintaining the same regret bound.
- For adversarial constraints and the dynamic baseline, RECOO achieves $O\left(P_T \sqrt{T}\right)$ regret and $O(T^{3/4})$ violation, which are the first regret and violation bounds in this setting.

| Reference | Loss Function | Regret, Violation | Dynamic Regret, Violation |
|---|---|---|---|
| [35] | | $O(\sqrt{T}), O(T^{3/4})$ | N/A |
| [30] | Convex | $O(\sqrt{T}), O(T^{1/4})$ | $O(\sqrt{P_T T}), O(\sqrt{T})$ |
| RECOO | | $O(\sqrt{T}), O(1)$ | $O(\sqrt{P_T T}), O(\log T)$ |
| [35] | | $O(\log T), O(\sqrt{T \log T})$ | N/A |
| [30] | Strongly Convex | $O(\log T), O(\log T)$ | $O(\sqrt{P_T T}), O(\sqrt{T})$ |
| RECOO | | $O(\log T), O(1)$ | $O(\sqrt{P_T T}), O(\log T)$ |

Table 1: Our results and related work for fixed constraints.

| Reference | Loss Function | Regret , Violation | Dynamic Regret , Violation |
|---|---|---|---|
| [31] | | $O(\sqrt{T}) , O(T^{3/4})$ | N/A |
| RECOO | Convex | $O(\sqrt{T}) , O(T^{3/4})$ | $O(P_T \sqrt{T}) , O(T^{3/4})$ |
| [31] | | $O(T^{c/2}) , O(T^{1-c/2})$ | N/A |
| RECOO | Strongly Convex | $O(\log T) , O(\sqrt{T \log T})$ | $O(P_T \sqrt{T}) , O(\sqrt{T \log T})$ |

Table 2: Our results and related work for adversarial constraints.

**Analysis:** The key idea behind these results is the rectified penalty imposed when making decisions, which enable us to develop a simple yet effective self-bounding relation that quantifies the trade-off between the regret and the cumulative hard constraints. The analysis is tailored for the cumulative hard constraints and is different from the drift-plus-penalty method used in [30, 31, 34], where the constraint violations have been established by bounding the dual variables (also called virtual queues).

## 1.2 Related work

COCO has been studied in the literature [17, 25, 20, 16, 6, 22, 35, 30, 31]. We have already presented a detailed comparison with the existing results on COCO-Hard. We next review some of recent results on COCO-Soft, COCO with soft constraints.

**COCO-Soft with fixed constraints:** [17] studies COCO-Soft and proposed an algorithm that achieves $O(\sqrt{T})$ regret and $O(T^{3/4})$ violation. This result has been extended in [14] to characterize the regret-violation tradeoff, in particular, the proposed algorithm achieves $O(T^{\max\{c,1-c\}})$ regret and $O(T^{1-c/2})$ violation, where $c$ controls the trade-off. Assuming Slater's condition holds, the algorithm in [34] achieves $O(\sqrt{T})$ regret and $O(1)$ violation and is designed based on a powerful "drift-plus-penalty" method, which inspires some of the design in RECOO.

**COCO-Soft with adversarial constraints:** Adversarial constraints are more difficult to satisfy but have been considered in the literature [25, 20, 16, 6, 22]. For COCO-Soft with adversarial constraints, the authors in [25] developed an online mirrored descent type algorithm that achieves $O(\sqrt{T})$ regret and $O(T^{3/4})$ violation. Later, [6, 16] generalized the baseline in [25] and still achieve $O(\sqrt{T})$ regret and $O(T^{3/4})$ violation. With Slater's condition, [20] presents an online gradient descent algorithm based on the drift-plus-penalty method [19], which achieves $O(\sqrt{T})$ regret and $O(\sqrt{T})$ violation. [22] extended the result to an online optimization with sub-modular losses. Note the key to reducing the constraint violation in these works is a refined bound on virtual queues (or dual variables) using the Lyapunov drift method under Slater's condition. It remains open that whether $O(T^{3/4})$ violation can be reduced with adversarial constraints (soft or hard) while keeping $O(\sqrt{T})$ regret *without* Slater's condition.

## 2 COCO-Hard

In this section, we formally define COCO-Hard. Consider the following online convex optimization problem: At each round $t \in [T]$, the learner makes decision $x_t$ and then observes the loss $f_t(x_t)$ and constraint function $g_t(\cdot)$ after the decision is executed. The goal of the learner is to generate a decision sequence $\{x_1, x_2, \ldots, x_{T-1}, x_T\}$ to minimize the total loss $\sum_{t=1}^{T} f_t(x_t)$ and the hard constraint violation $\sum_{t=1}^{T} g_t^+(x_t)$.

To quantify the performance of an online convex optimization algorithm for COCO-Hard, we will first compare it with a static baseline, called the best fixed decision in hindsight, which is the solution to the following offline COCO.

**Offline COCO**  The offline COCO is formulated as follows

$$\min_{x \in \mathcal{X}} \sum_{t=1}^{T} f_t(x) \text{ subject to: } g_t(x) \leqslant 0, \forall 1 \leqslant t \leqslant T, \tag{2}$$

where $\mathcal{X}$ is a "simple" convex set (e.g., positive quadrant or probability simplex), $\{f_t(\cdot)\}_t$ are convex loss functions and $\{g_t(\cdot)\}_t$ are convex constraint functions. The optimal solution $x^*$ to offline COCO is called the best single decision in hindsight, a widely used baseline for COCO [10, 17, 25, 32].

Based on offline COCO, the regret and constraint violation of an online algorithm are defined below.

**Regret and cumulative hard constraint violation (or violation for short)**

$$\mathcal{R}(T) := \sum_{t=1}^{T} f_t(x_t) - \sum_{t=1}^{T} f_t(x^*), \tag{3}$$

$$\mathcal{V}(T) := \sum_{t=1}^{T} g_t^+(x_t). \tag{4}$$

## 3 RECOO: A rectified online optimization algorithm

We present RECOO, an online optimization algorithm with the rectified decision and penalty update.

---

**RECOO — A Rectified Online Optimization Algorithm**

---

**Initialization:** $x_0 \in \mathcal{X}, Q(0) = 0, f_0(x) = g_0(x) = 0, \forall x \in \mathcal{X}$ and learning rates $\alpha_t, \eta_t, \gamma_t$.

For $t = 1, \cdots, T$,

- **Set:** $\hat{g}_{t-1}^+(x) = \gamma_{t-1} g_{t-1}^+(x)$.
- **Rectified decision:** find the optimal solution of $x_t$ by solving:

$$x_t = \arg\min_{x \in \mathcal{X}} \langle \nabla f_{t-1}(x_{t-1}), x - x_{t-1} \rangle + Q(t-1)\hat{g}_{t-1}^+(x) + \alpha_{t-1}\|x - x_{t-1}\|^2$$

- **Observe:** value $\nabla f_t(x_t)$ and constraint function $g_t(\cdot)$.
- **Rectified penalty update:** update the estimates of penalty variables $Q(t)$ as follows:

$$Q(t) = \max\left(Q(t-1) + \hat{g}_t^+(x_t), \eta_t\right).$$

---

In RECOO, the learning rates $\alpha_t, \eta_t, \gamma_t$ are chosen differently depending on whether the loss function is convex or strongly convex, but are oblivious to the type of constraints (fixed or adversarial). The choices of learning rates can be found in the theorem statements.

We explain the intuition behind RECOO and the importance of the "rectifiers". The Lagrange function of the offline optimization problem (2) is defined to be

$$L(\lambda, x) := \sum_{t=1}^{T} L_t(\lambda_t, x) := \sum_{t=1}^{T} f_t(x) + \lambda_t g_t(x),$$

where $\{\lambda_t\}$ are the dual variables associated with the constraints in (2).

Since we have no prior knowledge of $f_t(\cdot)$ when making decision $x_t$, we estimate it with the first-order approximation at $x_{t-1}$ based on the historical information as follows

$$\hat{f}_t(x) = f_{t-1}(x_{t-1}) + \langle \nabla f_{t-1}(x_{t-1}), x - x_{t-1} \rangle.$$

Moreover, we replace the original constraint function $g_t(\cdot)$ with $\hat{g}_{t-1}^+(\cdot)$ and the dual variables $\lambda_t$ with $Q(t-1)$ such that $Q(t-1)\hat{g}_{t-1}^+(x)$ is a rectified approximator of $\lambda_t g_t(x)$. We also added the the regularization (or smooth) term $\alpha_t\|x - x_t\|^2$ that helps the stability of the algorithm. Note this design is related to penalty-based proximal optimization where we aim to minimize an approximated $f_t(x)$ w.r.t. proximal operator on the "old" rectified function $\hat{g}_{t-1}^+(x)$.

In the penalty update of $Q(t)$, we first rectify the original constraint function $g_t(\cdot)$ with $\hat{g}_t^+(\cdot)$ and add it to $Q(t-1)$ such that penalty increases when violation occurs in each round. Further we rectify $Q(t)$ with a round-dependent constant $\eta_t$ to impose a "minimum" penalty price. The design of rectified penalty update induces conservative decisions in the decision-making step to minimize constraint violation. Note that it is different with the traditional primal-dual algorithm that does not rectify the constraint violation and impose a minimum penalty price, and when the price (dual variable) is zero, the algorithm can take very aggressive decisions that lead to large hard violation. This is not a problem when primal-dual algorithm is used as a numerical method for solving a constrained optimization problem, but leads to overly aggressive decisions and large violation when applying it to COCO. For a similar reason, RECOO rectifies the amount of violation $g_t(x)$ in the decision-making step so the violation does not become negative to prevent overly optimistic decisions. We will see that the rectifiers in both decision-making and penalty update leads to an upper bound on "regret + violation" as a whole (Lemma 1), which further leads separate upper bounds on regret and (hard) constraint violation. This approach is different from the primal-dual optimization that quantifies the constraint violation indirectly by bounding the dual variables/virtual queues.

Finally, we comment that our algorithm only needs to solve an "almost" unconstrained optimization problem ($\mathcal{X}$ is usually a simple set like the box constraints). Therefore, we might find its close-form with the inverse operation of the function by taking "gradient = zero" or gradient-based methods are sufficient to find the minimizer. For example, we could use the accelerated proximal-gradient method in [23] to find the minimizer with a linear and dimension-free converge rate. Our algorithm is more efficient than the projection online gradient descent algorithm [38] and online Frank-Wolfe algorithm [12], and has similar complexity with [30, 34]. Moreover, our algorithm does not assume any information on the type of constraints apriori and projection-based methods are only feasible when the constraint set is available beforehand.

We next analyze the regret and violation of RECOO as defined in (3) and (4) based on the following standard technical assumptions on the feasible set, loss and constraint functions.

**Assumption 1** *The feasible set $\mathcal{X}$ is convex with diameter $D$ such that $\|x - x'\| \leqslant D, \forall x, x' \in \mathcal{X}$.*

**Assumption 2** *The loss function is convex and Lipschitz continuous with Lipschitz constant $F$ such that $|f_t(x) - f_t(x')| \leqslant F\|x - x'\|, \forall x, x' \in \mathcal{X}, \forall t$.*

**Assumption 3** *The constraint function is convex and Lipschitz continuous with Lipschitz constant $G$ such that $|g_t(x) - g_t(x')| \leqslant G\|x - x'\|, \forall x, x' \in \mathcal{X}, \forall t$. For ease of exposition, assume $g_1(0) \leqslant 0$.*

Note Assumption 1 implies the feasible set $\mathcal{X}$ is bounded. Assumptions 2 and 3 mean the loss and constraint functions have bounded gradients. With these assumptions, we are ready to present our main result of RECOO for both fixed and adversarial constraints.

**Theorem 1** *Choose $\alpha_t = \sqrt{t}$, $\eta_t = \sqrt{t}$, and $\gamma_t = t^{\frac{1}{2}+\varepsilon}$, where $\varepsilon > 0$. Under Assumptions 1-3, RECOO algorithm achieves the following regret and constraint violation bounds:*

$$\mathcal{R}(T) \leqslant \left(\frac{F^2}{4} + D^2\right)\sqrt{T} \text{ (fixed or adversarial constraints),}$$

$$\mathcal{V}(T) \leqslant F^2 + FD\left(1 + \frac{1}{\varepsilon}\right) + D^2 \text{ (fixed constraints), and}$$

$$\mathcal{V}(T) \leqslant \left(F^2 + \frac{G^2}{4} + FD\left(5 + \frac{1}{\varepsilon}\right) + 2D^2\right)T^{\frac{3}{4}} \text{ (adversarial constraints).}$$

**Remark 1** *For fixed constraints, Theorem 1 establishes the optimal order-wise results of $O(\sqrt{T})$ regret and $O(1)$ violation since $\Omega(\sqrt{T})$ regret is the well-known lower bound for OCO with general convex loss and $O(1)$ is the best one can have. For adversarial constraints, by carefully choosing the learning rates in RECOO, we can establish a trade-off of $O(T^c)$ regret and $O(T^{1-c/2})$ cumulative violation where $c \in [1/2, 1)$ (see details of Corollary 1 in Appendix C). Moreover, Theorem 1 suggests that a large $\epsilon$ may imply small upper bounds of constraint violation. Our experiment results in Section 4 show that a small $\epsilon$ (e.g., $0.01$ or $0.1$) is sufficient to keep the violation small.*

Next, we show Theorem 1 can be further improved when the loss function is strongly convex.

**Assumption 4** *The loss functions $f_t(\cdot), \forall t$, are $\mu$-strongly convex in $\mathcal{X}$ for $\mu > 0$, i.e. $f_t(x') \geqslant f_t(x) + \langle x' - x, \nabla f_t(x)\rangle + \frac{\mu}{2}\|x - x'\|^2, \forall x, x' \in \mathcal{X}, \forall t$.*

With Assumption 4, we are able to establish a $O(\log T)$ regret whiling keeping the violation bounds at the same or a smaller order.

**Theorem 2** *Choose the learning rates to be $\alpha_t = \frac{\mu t}{2}, \eta_t = \sqrt{t}, \gamma_t = t^{\frac{1}{2}+\varepsilon}, \forall t \in [T]$, where $\varepsilon > 0$. Under Assumptions 1-4, RECOO achieves the following regret and violation bounds:*

$$\mathcal{R}(T) \leqslant \frac{F^2}{2\mu}(1 + \log T) \text{ (fixed or adversarial constraints),}$$

$$\mathcal{V}(T) \leqslant \frac{F^2}{\mu} + FD\left(1 + \frac{1}{\epsilon}\right) \text{ (fixed constraints), and}$$

$$\mathcal{V}(T) \leqslant \left(\frac{F^2}{\mu} + \frac{G^2}{4} + FD\left(1 + \frac{1}{\varepsilon} + \frac{4}{\mu}\right) + D^2\right)\sqrt{T(1 + \log T)} \text{ (adversarial constraints).}$$

**Remark 2** *The $O(\log T)$ regret is a well-known result in OCO (unconstrained) for a strongly convex loss function [11]. Therefore, for fixed constraints, our results are order-wise sharp because $O(1)$ violation is the best one can achieve.*

**Extension to a dynamic baseline**

So far, the main results are against a static baseline $x_t^* = x^*, \forall t$. We next analyze the performance of RECOO against a dynamic baseline as in [37, 30], where $x_{t+1}^*$ could be different from $x_t^*$ but with limited variation such that $\sum_{t=1}^{T-1}\|x_{t+1}^* - x_t^*\| \leqslant P_T$. We define the regret against the dynamic baseline to be $\mathcal{R}^{\text{dynamic}}(T) := \sum_{t=1}^{T} f_t(x_t) - \sum_{t=1}^{T} f_t(x_t^*)$. Without any modifications, we show that RECOO algorithm can achieve $O(\sqrt{T(1 + P_T)})$ regret and $O(1)$ violation in the following theorem.

**Theorem 3** *Under Assumptions 1-3, let the learning rates be $\alpha_t = \sqrt{t}, \eta_t = \sqrt{t}, \gamma_t = t^{\frac{1}{2}+\varepsilon}, \forall t \in [T]$, where $\varepsilon > 0$. Let $\{x_t^*\}$ be the optimal solution to (2) with an additional constraint that $\sum_{t=1}^{T-1} \|x_{t+1} - x_t\| \leqslant P_T$. RECOO achieves the following bounds on the regret and cumulative constraint violations:*

$$\mathcal{R}^{dynamic}(T) \leqslant \left(\frac{F^2}{2} + D^2 + 2DP_T\right)\sqrt{T+1} \quad \text{(fixed or adversarial constraints)},$$

$$\mathcal{V}(T) \leqslant F^2 + FD\left(1 + \frac{1}{\varepsilon}\right) + D^2 \quad \text{(fixed constraints), and}$$

$$\mathcal{V}(T) \leqslant \left(F^2 + \frac{G^2}{4} + FD\left(5 + \frac{1}{\varepsilon}\right) + 2D^2\right)T^{\frac{3}{4}} \quad \text{(adversarial constraints)}.$$

**Remark 3** *RECOO achieves $O\left(P_T\sqrt{T}\right)$ regret and $O(1)$ violation for fixed constraints, which improves $O\left(P_T\sqrt{T}\right)$ regret and $O(T^{1/4})$ violation in [30]. Moreover, RECOO achieves $O\left(P_T\sqrt{T}\right)$ regret and $O(T^{3/4})$ violation for adversarial constraints, which is a new result. As shown in Corollary 2 in Appendix F, combining RECOO with expert-tracking proposed in [37], we achieve $O\left(\sqrt{P_TT}\right)$ regret and $O(\log T)$ violation for fixed constraints, which improves $O\left(\sqrt{P_TT}\right)$ regret and $O(\sqrt{T})$ violation in [30].*

Next, we provide the detailed analysis of Theorem 1. The proofs of Theorem 2 and Theorem 3 can be found in Appendix.

### 3.1 Proof of Theorem 1

We first introduce a key lemma that establishes an upper bound on "regret plus violation" at each round $t$, which we called self-bounding property.

**Lemma 1 (Self-Bounding Property)** *Let $x$ be any feasible solution to offline COCO (2) and $x_t$ be the optimal solution returned by the RECOO algorithm. We have*

$$f_t(x_t) - f_t(x) + Q(t)\hat{g}_t^+(x_{t+1}) \leqslant \frac{F^2}{4\alpha_t} + \alpha_t\|x - x_t\|^2 - \alpha_t\|x - x_{t+1}\|^2 \tag{5}$$

Define $x^*$ to be the optimal solution to offline COCO in (2) and let $x = x^*$ in Lemma 1, we establish the regret and the constant violation bounds in Theorem 1.

**Proof of Theorem 1: fixed constraints**

Note that $Q(t)\hat{g}_t^+(x_{t+1})$ is nonnegative. Based on (5), we have

$$f_t(x_t) - f_t(x^*) \leqslant \frac{F^2}{4\alpha_t} + \alpha_t\|x^* - x_t\|^2 - \alpha_t\|x^* - x_{t+1}\|^2, \tag{6}$$

$$Q(t)\hat{g}_t^+(x_{t+1}) \leqslant \frac{F^2}{4\alpha_t} + |f_t(x_t) - f_t(x^*)| + \alpha_t\|x^* - x_t\|^2 - \alpha_t\|x^* - x_{t+1}\|^2, \tag{7}$$

which we will use next to establish the bounds on regret and violation.

**Regret bound:** We sum up inequality (6) for $t = 1, \cdots, T$ and have

$$\sum_{t=1}^{T}(f_t(x_t) - f_t(x^*)) \leqslant \frac{F^2}{4}\sum_{t=1}^{T}\frac{1}{\alpha_t} + \sum_{t=1}^{T}(\alpha_t - \alpha_{t-1})\|x^* - x_t\|^2$$

$$\leqslant \frac{F^2}{4}\sum_{t=1}^{T}\frac{1}{\alpha_t} + D^2\sum_{t=1}^{T}(\alpha_t - \alpha_{t-1})$$

where the first inequality holds by dropping the last negative term and the last inequality holds by Assumption 1. Choose $\alpha_t = \sqrt{t}$, we have

$$\sum_{t=1}^{T}(f_t(x_t) - f_t(x^*)) \leqslant \left(\frac{F^2}{2} + D^2\right)\sqrt{T}.$$

**Violation bound:** We substitute the definition of $\hat{g}_t^+(\cdot)$ in (7) and have

$$g_t^+(x_{t+1}) \leqslant \frac{F^2}{4Q(t)\alpha_t\gamma_t} + \frac{|f_t(x_t) - f_t(x^*)|}{Q(t)\gamma_t} + \frac{\alpha_t}{Q(t)\gamma_t}\|x_t - x^*\|^2 - \frac{\alpha_t}{Q(t)\gamma_t}\|x_{t+1} - x^*\|^2$$

$$\leqslant \frac{F^2}{4t^{\frac{3}{2}+\varepsilon}} + \frac{|f_t(x_t) - f_t(x^*)|}{t^{1+\varepsilon}} + \frac{\|x_t - x^*\|^2 - \|x_{t+1} - x^*\|^2}{t^{\frac{1}{2}+\varepsilon}},$$

where the second inequality holds by $Q(t) \geqslant \eta_t$ according to the rectified penalty update $Q(t) = \max\left(Q(t-1) + \hat{g}_t^+(x_t), \eta_t\right)$. The inequality implies that

$$\sum_{t=1}^{T} g_t^+(x_{t+1}) \leqslant \sum_{t=1}^{T} \frac{F^2}{4t^{\frac{3}{2}+\varepsilon}} + \sum_{t=1}^{T} \frac{|f_t(x_t) - f_t(x^*)|}{t^{1+\varepsilon}} + \sum_{t=1}^{T} \frac{\|x_t - x^*\|^2 - \|x_{t+1} - x^*\|^2}{t^{\frac{1}{2}+\varepsilon}}$$

$$\leqslant F^2 + FD + \frac{FD}{\varepsilon} + D^2 \tag{8}$$

where the second inequality holds by Lemma 6, which includes the detailed calculations of the three terms above. Note that the left-hand-side of inequality (8) is not the violation because the index mismatch (the violation at round $t$ is $g_t^+(x_t)$ not $g_t^+(x_{t+1})$).

For fixed constraints, i.e. $g_t(x) = g(x)$, the inequality above implies that $\mathcal{V}(T) := \sum_{t=1}^{T} g^+(x_t) \leqslant F^2 + FD\left(1 + \frac{1}{\varepsilon}\right) + D^2$. We have proved the first part of Theorem 1 for the fixed constraints. Let us continue with (8) to prove the second part of Theorem 1 for adversarial constraints.

**Proof of Theorem 1: adversarial constraints**

To quantify $\mathcal{V}(T) := \sum_{t=1}^{T} g_t^+(x_t)$ for the adversarial constraints, we need to establish the relationship between $g_t^+(x_t)$ and $g_t^+(x_{t+1})$ to address the index mismatch, which can be bounded by $\|x_{t+1} - x_t\|^2$ as shown in the following lemma.

**Lemma 2** *Under Assumptions 1-3, RECOO achieves for any $\beta > 0$*

$$g_t^+(x_t) - g_t^+(x_{t+1}) \leqslant \frac{G^2}{4\beta} + \beta\|x_t - x_{t+1}\|^2.$$

The next lemma further quantifies $\sum_{t=1}^{T}\|x_{t+1} - x_t\|^2$.

**Lemma 3** *Under Assumptions 1-3, RECOO achieves*

$$\sum_{t=1}^{T}\|x_{t+1} - x_t\|^2 \leqslant 4FD\sqrt{T} + D^2.$$

Take summation of this equality in Lemma 2 from $t = 1$ to $T$, and we have

$$\sum_{t=1}^{T}\left(g_t^+(x_t) - g_t^+(x_{t+1})\right) \leqslant \frac{G^2 T}{4\beta} + \beta\sum_{t=1}^{T}\|x_t - x_{t+1}\|^2$$

$$\leqslant \left(\frac{G^2}{4} + 4FD\right)T^{3/4} + D^2 T^{1/4}, \tag{9}$$

where the last inequality holds by letting $\beta = T^{1/4}$. Combining (8) and (9) completes the proof.

## 4 Experiments

In this section, we present synthetic and real dataset experiments for evaluating the performance of RECOO with fixed and adversarial constraints. We compared RECOO with the algorithms in [35, 30] for the fixed-constraints setting; with the algorithm in [31] for the adversarial-constraints setting; and with the algorithms in [31, 32] for the real-dataset setting. All results are obtained by averaging over 500 trials and reported with 95% confidence interval.

**Fixed constraints** Similar to [30], we considered loss functions $f_t(x) = \langle \theta(t), x \rangle$, where $\theta(t)$ is time varying and unknown at round $t$; and the fixed constraint function $Ax \leqslant b$, where $x \in \mathbb{R}^2$,

$A \in \mathbb{R}^{3 \times 2}$ and $b \in \mathbb{R}^3$. We chose the number of total rounds to be $T = 5,000$, and chose $\theta(t) = [\theta_1(t), \theta_2(t), \theta_3(t)]$, where $\theta_1(t) \sim U(-t^{1/10}, +t^{1/10})$; $\theta_2(t) \sim U(-1, 0)$ when $t \in [1, 1500] \cup [2000, 3500] \cup [4000, 5000]$ and $\theta_2(t) \in U(0, 1)$ otherwise; and $\theta_3(t) = (-1)^{\mu(t)}$ where $\mu(t)$ is a random permutation of vector $[1 : 5000]$. Let $\mathcal{X} = \{x | 0 \leqslant x_1 \leqslant 1, 0 \leqslant x_2 \leqslant 1\}$, $A_{i,j} \sim U(0.1, 0.5), 1 \leqslant i \leqslant 3, 1 \leqslant j \leqslant 2$, and $b_i \sim U(0, 0.3), 1 \leqslant i \leqslant 3$, respectively.

We compared RECOO with the algorithm in [35] and Algorithm 1 and Algorithm 2 in [30], where the learning rates are summarized in Table 3 in Appendix G, which are the values suggested in [35] and [30]. Figure 1a illustrates the trajectories of the cumulative loss $\sum_{t=1}^{T} f_t(x_t)$, where we observe that all algorithms have similar trends while RECOO achieves the best cumulative loss. Figure 1b shows RECOO achieves the smallest cumulative violation. In particular, the (mean, variance) pair of RECOO for loss and violation are $(-510.23, 32.78)$ and $(0.05, 0.008)$ at the end of the learning horizon; that of Algorithm 1 in [30] are $(-507.17, 36.46)$ and $(0.23, 0.007)$; that of Algorithm 2 in [30] are $(-509.09, 36.73)$ and $(0.16, 0.008)$; and that of Algorithm 1 in [35] are $(-504.81, 36.82)$ and $(980.64, 10.49)$. These results and Figure 1 confirm that RECOO outperforms the existing algorithms w.r.t. both cumulative loss and violation.

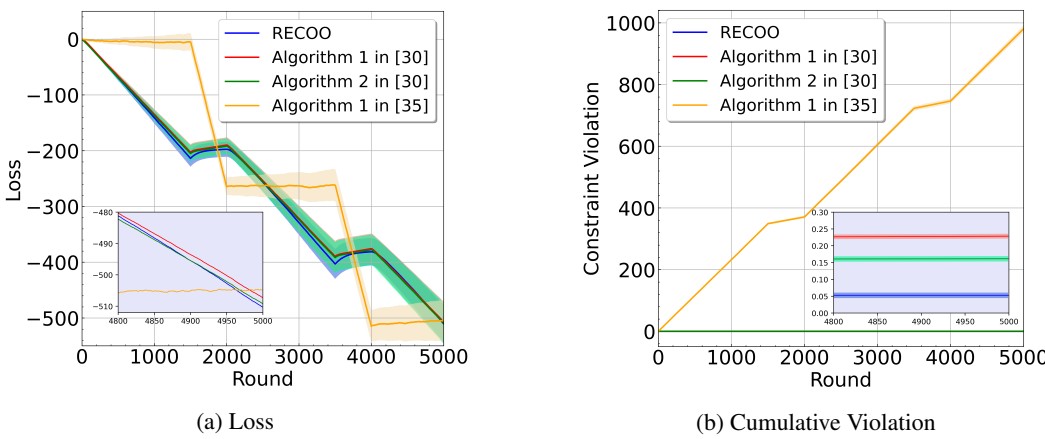

(a) Loss          (b) Cumulative Violation

Figure 1: Experiment with fixed constraints

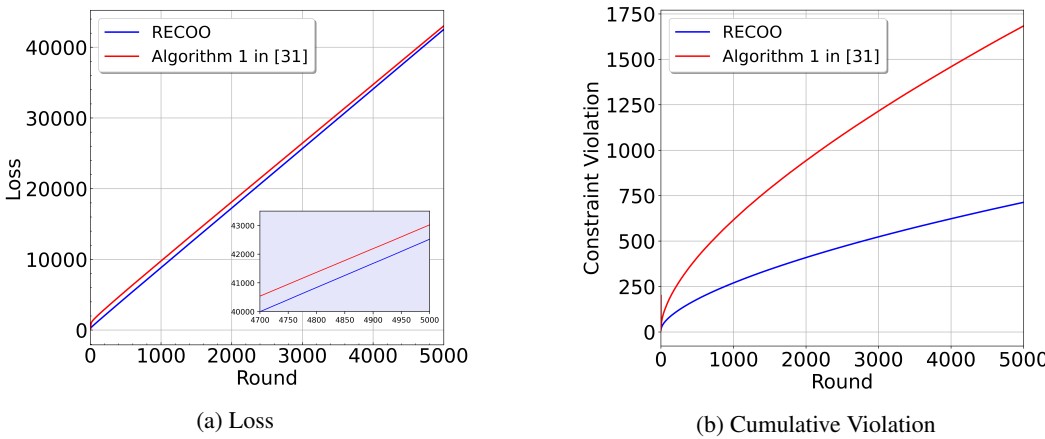

(a) Loss          (b) Cumulative Violation

Figure 2: Experiment with adversarial constraints

**Experiment with adversarial constraints** We considered an OCO with adversarial constraints similar as the one in [31]. The loss function was chosen to be $f_t(x) = \frac{1}{2} \| H(t) \cdot x - y(t) \|^2$ with $H(t) \in R^{4 \times 10}$, $x \in \mathbb{R}^{10}$ and $y(t) \in \mathbb{R}^4$ where $H_{i,j}(t) \sim U(-1, 1), 1 \leqslant i \leqslant 4, 1 \leqslant j \leqslant 10$, and $y_i(t) = \sum_{j=1}^{i,j} H_{i,j}(t) + \varepsilon_i$ with $\varepsilon_i$ being the standard normal random variable for any $i$. The constraint functions were chosen to be $g_t(x) = A(t)x - b(t)$ with $A(t) \in \mathbb{R}^{2 \times 10}$ and $b(t) \in \mathbb{R}^2$, where $A_{i,j}(t) \sim U(0, 2), \forall i, j, t$ and $b_i(t) \sim U(0, 1), \forall i, t$, respectively.

We compared RECOO with Algorithm 1 in [31], where the learning rates are summarized in Table 4 in Appendix G. Figure 2 includes the cumulative losses and violations. In particular, the (mean, variance) pair of RECOO for loss and violation are $(42510.14, 38.05)$ and $(713.45, 1.60)$ at the end of learning horizon, respectively while that of Algorithm 1 in [31] are $(43011.29, 56.17)$ and $(1684.17, 2.85)$. Therefore, RECOO performs better, especially in terms of the cumulative violation, which further shows the "rectified" design reduces the cumulative hard constraint violation.

**Experiment of online job scheduling in distributed data centers** We also tested our algorithm for online job scheduling in a distributed data center similar as in [32]. We considered a distributed data center with server clusters located at different regions. The incoming jobs arrive at a front-end load balancer and will be scheduled to different clusters to fulfill the service. The service capability of a cluster is a function w.r.t. its energy consumption and the energy prices vary across locations and times. The goal is to minimize the energy cost while guaranteeing real-time processing of safety-critical jobs. This problem can be formulated as a constrained online convex optimization problem and solved by our algorithm.

Specifically, we considered $r = 10$ regions, each region has 10 clusters, and each time/round $t$ has 5 minutes. Let $x_t \in \mathbb{R}^{100}$ be the energy allocation vector of all clusters at round $t$, where the $i$th entry is the energy allocation of cluster $i$. Let $f_t(x_t) = \langle c_t, x_t \rangle$, where $c_t \in \mathbb{R}^{100}$ are the energy prices at time $t$. Let $g_t(x_t) = \lambda_t - \sum_{i=1}^{100} h_i(x_{t,i})$, where $\lambda_t$ is the number of job arrivals during time $t$ and $h_i(x_{t,i}) = 4\log(1 + 4x_{t,i})$ is the service capacity of cluster $i$ at round $t$. The constraint violation represents the number of delayed jobs (jobs not severed in real-time). In the experiment, we used the electricity price trace (i.e., $\{c_t\}$) between $05/01/2017$ and $05/10/2017$ at 10 different regions in New York city from NYISO [1]. We calibrated job arrivals of a realistic traffic pattern from [28] with a non-stationary Poisson process $\{\lambda_t\}$ to replace the stationary traffic studied in [32].

We compared our algorithm with [31] and [32], where learning rates are summarized in Table 5 in Appendix G. We plotted average energy costs and constraint violations in Figure 3. It shows that our algorithm achieves better performance on the loss and constraint violation (the number of delayed jobs) compared to [31] and [32].

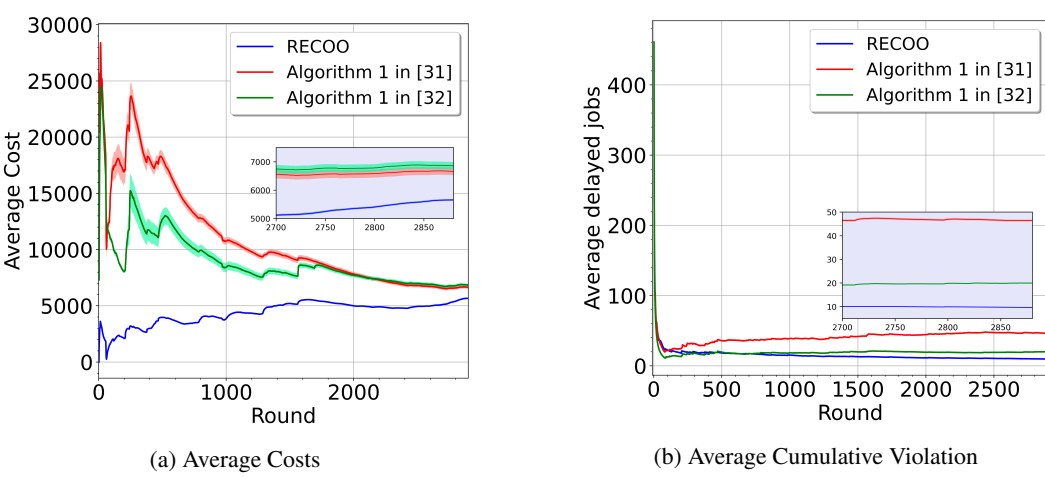

(a) Average Costs  (b) Average Cumulative Violation

Figure 3: Experiment of online job scheduling in a distributed data center

## 5 Conclusions

In this paper, we studied online convex optimization with hard constraints (COCO-Hard) under two settings: fixed constraints and adversarial constraints. We proposed a RECtified Online Optimization algorithm (RECOO) and proved it achieves the best of two worlds and beyond. The algorithm improves the best-known results for the fixed-constraints, matches the best results for the adversarial-constraints, outperforms the state-of-art results when the loss functions are strongly-convex. The experiments confirmed our theoretical results.

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
