# OpenReview forum: "Online Convex Optimization with Hard Constraints: Towards the Best of Two Worlds and Beyond"
_NeurIPS.cc/2022/Conference — NeurIPS 2022 Accept_

### Official Review · Reviewer_YFq8 · 2022-07-04

**Rating:** 6
**Confidence:** 5
**Soundness:** 3 good
**Presentation:** 3 good
**Contribution:** 3 good

**Summary:**

This paper studies an extension of the well-known Online Convex Optimization (OCO) problem where at each round $t\in[T]$, upon committing to an action $x_t$, both a loss function $f_t$ and also a constraint function $g_t$ are revealed. The goal is to minimize the overall incurred loss $\sum_{t=1}^T f_t(x_t)$ while minimizing the hard constraint violation $\sum_{t=1}^T \max (g_t(x_t),0)$. The authors propose a single algorithm called RECOO that obtains state-of-art bounds for static/dynamic regrets and constraint violation for convex/strongly convex losses. The numerical experiments verify the theoretical findings as well.

**Questions:**

I have already mentioned many of my suggestions and questions earlier. In addition:
- Does the expert-tracking technique of [33] for obtaining optimal $O(\sqrt{P_T T})$ regret bounds apply to the setting with adversarial constraints as well? (in the paper, you have only applied to the setting with fixed constraints)
- The proofs provided in the text are not very insightful. They could be deferred to the appendix and the space could be used for additional experiments with real-world datasets.
-  In the statement of Theorem 1 and Theorem 3, state that the loss functions are assumed to be convex. It is currently missing.

**Limitations:**

The scope of their proposed algorithm has been clearly specified. This theoretical work does not have any potential negative societal impacts and there is no need for addressing it.

**Strengths And Weaknesses:**

Strengths:
- Paper is well-written and the authors have done an excellent job motivating the problem and giving intuitions for their algorithm.
- The literature review is comprehensive and clearly compares and contrasts related works to this paper.
- Imposing a time-varying minimum penalty price in the update for $Q(t)$ seems novel and changes the proof techniques for bounding the constraint violation.

Weaknesses:
- While the experiments clearly highlight the theoretical contributions of the paper, they have been done only for synthetic datasets. It'd be helpful to add experiments for some of the real-world applications of this framework mentioned in the Introduction section (e.g., safety-critical applications).
- The claim that the algorithm manages to achieve the "best of two worlds" is a bit misleading. "Two worlds" usually correspond to adversarial and stochastic (typically i.i.d.) constraints, however, in this paper, it refers to adversarial and fixed constraints.
- In the paper, it needs to be clarified what Slater's condition is, how so many of the prior works have assumed this condition holds to obtain better bounds, and how the framework in this paper does not make such an assumption.
- The notion of path length $P_T$ has been used several times in the paper before it is finally defined on page 6.
- The natural static benchmark for this problem should only satisfy the constraint $\sum_{t=1}^T \max (g_t(x),0)$, however, in the paper, the benchmark is further restricted to satisfy $g_t(x)\leq 0~\forall t\in[T]$. The authors should explain and motivate this choice of benchmark, and mention any potential hardness results for regret against the more natural static benchmark.
- The algorithm is quite similar to that of [18] and [30] (putting aside the new idea of rectifying $Q(t)$). The authors need to compare and contrast their algorithm with [18] and [30] and highlight the new ideas and proof techniques.

---

> ### Author Response · Authors · 2022-08-02
> **Response to Reviewer YFq8**
>
> We sincerely thank the reviewer for the encouraging and constructive comments. We'd like to incorporate these great suggestions in our revision and response the major concern in the following.
>
> **Experiment of online job scheduling in a distributed data center:** We tested our algorithm with the experiment of online job scheduling in a distributed data center similar as in [28]. We study a distributed data center infrastructure with server clusters located at different regions. The incoming jobs arrive at a front-end load balancer and will be scheduled to different clusters to fulfill the service. The service capability of a cluster is a function w.r.t. its energy consumption, and the energy prices vary across locations and times. The goal is to minimize the energy cost while guaranteeing real-time processing for safety-critical jobs. This problem can be formulated as a constrained online convex optimization problem and solved by our algorithm as follows.
>
> Specifically, we consider $r=10$ regions, each region has 10 clusters, and each time/round $t$ has 5 minutes. Let $x_t\in \mathbb R^{100}$ be the energy allocation vector of all clusters at round $t$, where the $i$th entry is the energy allocation of cluster $i$. Let $f_t(x_t) = \langle c_t, x_t \rangle$, where $c_t \in \mathbb R^{100}$ are the energy prices at time $t$. Let $g_t(x_t) = \lambda_t - h(x_t)$, where $\lambda_t$ is the number of job arrivals during time $t$ and $h(x_t)$ is the total service capacity of the data center at time $t$. The constraint violation represents the number of delayed jobs (jobs not severed in real-time). In the experiment, we use a 10-days electricity price trace (i.e., $\{c_t\}$) by extracting 10 regions from New York city. We calibrated job arrivals of a realistic traffic pattern with a non-stationary Poisson process $\{\lambda_t\}$ to replace the stationary traffic studied in [28].
>
> We consider [27] and [28] in the paper as our baselines. We show the average energy costs (the first table) and constraint violation (the second table) w.r.t. time in the following. It is verified that our algorithm achieves better performance on the loss and constraint violation compared to [27] and [28]. Please find the figures (Figure 3) for a better view in Appendix G of our revision.
> |       $T$           |    500    |    1000   |   1500   |   2000   |   2500   |   2880   |
> |:---------------------:|:---------:|:---------:|:--------:|:--------:|:--------:|:--------:|
> |     Our Algorithm     |  3968.823 |  3854.507 | 5206.972 | 4998.419 | 4948.378 | 5657.571 |
> | Algorithm 1 in [27] | 18102.464 | 10764.099 | 9302.034 | 7818.307 | 6807.092 | 6662.214 |
> | Algorithm 1 in [28] | 12704.005 |  8430.739 | 7806.612 | 7723.638 | 6989.628 | 6860.589 |
>
> |       $T$          |   500  |  1000  |  1500  |  2000  |  2500  |  2880  |
> |:---------------------:|:------:|:------:|:------:|:------:|:------:|:------:|
> |     Our Algorithm     | 18.361 | 15.282 | 12.968 | 11.316 | 10.547 |  9.668 |
> | Algorithm 1 in [27] | 35.872 | 38.908 | 41.721 | 44.463 | 47.358 | 46.444 |
> | Algorithm 1 in [28] | 19.507 | 12.968 | 19.840 | 19.341 | 19.096 | 19.977 |
>
> **Clarification on Slater's condition:** The Slater's condition is *"there exists a positive $\epsilon>0$ and $x\in \mathcal X$ such that $g_t(x) \leq -\epsilon, \forall t \in [T]$"* The Slater's condition is necessary to improve the soft constraint violation in [18, 28, 30]. The key improvement in [18, 28, 30] is that the Lyapunov drift technique can provide a refined bound on virtual queues (or dual variables) with Slater's condition, which thus achieves a smaller soft constraint violation. The reason we did not make such an assumption is we can quantify the "regret + violation'' as a whole, from which we can establish the regret and (hard) constraint violation directly, which is unlike the Lyapunov drift methods in [18, 28, 30] that quantify the constraint violation indirectly by bounding the dual variables/virtual queues.
>
> **Proper benchmark:** The current benchmark is the best fixed decision in hindsight, which minimizes the total costs while satisfying the constraints at each round. It serves as a classical benchmark in (constrained) online convex optimization [9, 18, 28, 30] and has wide applications (e.g., prediction with expert advice or recommendation systems). The mentioned benchmark $\sum_{t=1}^T g^{+}_t(x) \leq C$ is interesting and more challenging. For the case of $C=0$, it reduces to the classical benchmark studied in the paper. For the cases of $C > 0$ or even $C$ is a function of $T$, the baseline is strong and adaptive, which potentially introduces large regret for any casual online learning algorithm. This baseline has not been investigated in the literature before and we will definitely look into this problem in the future.

---

> > ### Author Response · Authors · 2022-08-02
> > **Response to Reviewer YFq8**
> >
> > **Comparison to [18] and [30]:** The design principle and proof techniques of our algorithm are different from [18] and [30]. For the design principle, our algorithm leveraged the idea of penalty-based proximal optimization to handle the constraints directly; and [18] and [30] leveraged the idea of primal-dual optimization and used the dual variables/virtual queues as the proxy of the constraint violation. For the proof techniques, our techniques quantify the "regret + violation" as a whole and establish the regret and (hard) constraint violation directly, which is unlike the Lyapunov drift methods in [18] and [30] that quantify the constraint violation indirectly by bounding the dual variables/virtual queues.
> >
> > **Questions:**
> >
> > Q1) We tried to apply the expert-tracking technique in the adversarial constraints, and it did not improve the theoretical results. The main challenge is that the constraint violation could be uncontrollable large with the expert-tracking technique, which we currently cannot overcome. It definitely would be an interesting problem to be investigated further.
> >
> > Q2-Q3) We have added the missing assumptions in Assumptions 2 and 3, and will re-organize the paper according to your suggestions.
> >
> > --------------------------- reference in our previous version --------------------------------
> >
> > [9] Elad Hazan. Introduction to online convex optimization. Foundations and Trends® in Optimization, 2016.
> >
> > [18] Michael J Neely and Hao Yu. Online convex optimization with time-varying constraints. arXiv preprint arXiv:1702.04783, 2017
> >
> > [28] Hao Yu, Michael Neely, and Xiaohan Wei. Online convex optimization with stochastic constraints. Advances in Neural Information Processing Systems, 2017.
> >
> > [30] Hao Yu and Michael J. Neely. A low complexity algorithm with $O(\sqrt{T})$ regret and $O(1)$ constraint violations for online convex optimization with long term constraints. The Journal of Machine Learning Research, 2020.

---

### Official Review · Reviewer_bywb · 2022-07-09

**Rating:** 6
**Confidence:** 3
**Soundness:** 3 good
**Presentation:** 3 good
**Contribution:** 3 good

**Summary:**

Online convex optimization (OCO) is a significant part of online learning literature. In real-world applications, there are constraints that need to be satisfied through optimization. This paper theoretically analyzes the constraint OCO under fixed and adversarial settings when the loss is either convex or strongly convex. Unlike the previous works, it considers the hard constraint violation which means that the average constraint violation (CV) is the average of the positive constraint violation and ignores the negative one which can lead to a minima which after a large iteration gives an infeasible solution. It also considers two kinds of regret: 1- static regret and 2- dynamic regret. Under hard constraint assumption,  for all of these different settings, it either recovers the existing results for CV or improves it.


**Questions:**

1- In the experimental result, we see that the CV bound for the existing methods (Fig 1) also shows the constant bound and not sublinearly as claimed in Table 1 or 2. What do you think is the reason for this observation?


**Limitations:**

This works addresses the potential social impacts.


**Strengths And Weaknesses:**

Clarity

The paper is written clearly with enough explanation for different settings. It explains well the significance of the hard constraint setting. The proofs are presented in an understandable manner.

Since the paper mainly is theoretical, it is not cleared enough why it gets better CV bound w.r.t. existing results. For example, explain which technical method resulted in that bound.

Originality

Considering the hard constraint in OCO is novel and also important for real-world problems.

Quality

The submission is technically sound. I’ve checked the proofs and they look correct and sound. However, the technique they’ve leveraged are not new and have been used in previous works.

Significance

The result is important for the real-world problem and besides, they improve the CV bound in a few settings compared to the existing results.


Limitation

The experimental results are just for toy examples. It would illustrate the significance of the proposed method if you have some results on a real-world dataset.

---

> ### Author Response · Authors · 2022-08-02
> **Response to Reviewer bywb**
>
> We greatly appreciate the reviewers' constructive comments and positive evaluation towards the novelty of this paper. We'd like to response to the major comments in the following.
>
> **Explanation on why our algorithm achieves a better bound:** Intuitively, the reason that our algorithm achieves a better bound lies in the rectified design of $Q(t)$ that imposes a minimum penalty at each round such that our algorithm avoids using overly aggressive actions. Unlike the traditional primal-dual based methods [26-28] and [30], it might choose aggressive actions that violate the constraints when the ''dual penalty'' is small. Technically, with the rectified design of $Q(t)$ and the clipped constraint function $\hat g^{+}(\cdot)$, our algorithm provides an upper bound on the ''regret + violation'' at any time $t$, i.e., $f_t(x_t) - f_t(x^*) + Q(t)\hat g^{+}_t(\cdot)$. It quantifies the ''regret + violation'' as a whole and we are able to establish regret and (hard) constraint violation directly. These techniques again are different with the traditional methods in [26-28] and [30] that quantify the constraint violation indirectly by bounding the dual variables/virtual queues. We have emphasized the key technical contributions in the revision.
>
> **Experiment of online job scheduling in a distributed data center:** We tested our algorithm with the experiment of online job scheduling in a distributed data center similar as in [28]. We study a distributed data center infrastructure with server clusters located at different regions. The incoming jobs arrives at a front-end load balancer and will be scheduled to different clusters to fulfill the service. The service capability of a cluster is a function w.r.t. its energy consumption and the energy prices vary across locations and times. The goal is to minimize the energy cost while guaranteeing real-time processing for safety-critical jobs. This problem can be formulated as a constrained online convex optimization problem and solved by our algorithm as follows.
>
> Specifically, we consider $r=10$ regions, each region has 10 clusters, and each time/round $t$ has 5 minutes. Let $x_t\in \mathbb R^{100}$ be the energy allocation vector of all clusters at round $t$, where the $i$th entry is the energy allocation of cluster $i$. Let $f_t(x_t) = \langle c_t, x_t \rangle$, where $c_t \in \mathbb R^{100}$ are the energy prices at time $t$. Let $g_t(x_t) = \lambda_t - h(x_t)$, where $\lambda_t$ is the number of job arrivals during time $t$ and $h(x_t)$ is the total service capacity of the data center at time $t$. The constraint violation represents the number of delayed jobs (jobs not severed in real-time). In the experiment, we use a 10-days electricity price trace (i.e., $\{c_t\}$) by extracting 10 regions from New York city. We calibrated job arrivals of a realistic traffic pattern with a non-stationary Poisson process $\{\lambda_t\}$ to replace the stationary traffic studied in [28].
>
> We consider [27] and [28] in the paper as our baselines. We show the average energy costs (the first table) and constraint violation (the second table) w.r.t. time in the following. It is verified that our algorithm achieves better performance on the loss and constraint violation compared to [27] and [28]. Please find the figures (Figure 3) for a better view in Appendix G of our revision.
> |       $T$           |    500    |    1000   |   1500   |   2000   |   2500   |   2880   |
> |:---------------------:|:---------:|:---------:|:--------:|:--------:|:--------:|:--------:|
> |     Our Algorithm     |  3968.823 |  3854.507 | 5206.972 | 4998.419 | 4948.378 | 5657.571 |
> | Algorithm 1 in [27] | 18102.464 | 10764.099 | 9302.034 | 7818.307 | 6807.092 | 6662.214 |
> | Algorithm 1 in [28] | 12704.005 |  8430.739 | 7806.612 | 7723.638 | 6989.628 | 6860.589 |
>
>
> |       $T$          |   500  |  1000  |  1500  |  2000  |  2500  |  2880  |
> |:---------------------:|:------:|:------:|:------:|:------:|:------:|:------:|
> |     Our Algorithm     | 18.361 | 15.282 | 12.968 | 11.316 | 10.547 |  9.668 |
> | Algorithm 1 in [27] | 35.872 | 38.908 | 41.721 | 44.463 | 47.358 | 46.444 |
> | Algorithm 1 in [28] | 19.507 | 12.968 | 19.840 | 19.341 | 19.096 | 19.977 |
>
> **The possible reasons on the small violation in Figure 1:** The possible reasons could be the low dimension of decision variables and the relatively static and loose constraints, which make the baseline algorithms quickly adapt to the violation as well.

---

> > ### Author Response · Authors · 2022-08-02
> > **Response to Reviewer bywb**
> >
> > --------------------------- reference in our previous version --------------------------------
> >
> > [26] Xinlei Yi, Xiuxian Li, Tao Yang, Lihua Xie, Tianyou Chai, and Karl Johansson. Regret and cumulative constraint violation analysis for online convex optimization with long term constraints. In International Conference on Machine Learning, 2021.
> >
> > [27] Xinlei Yi, Xiuxian Li, Tao Yang, Lihua Xie, Tianyou Chai, and Karl H Johansson. Regret and cumulative constraint violation analysis for distributed online constrained convex optimization. arXiv preprint arXiv:2105.00321, 2021.
> >
> > [28] Hao Yu, Michael Neely, and Xiaohan Wei. Online convex optimization with stochastic constraints. Advances in Neural Information Processing Systems, 2017.
> >
> > [30] Hao Yu and Michael J. Neely. A low complexity algorithm with $O(\sqrt{T})$ regret and $O(1)$ constraint violations for online convex optimization with long term constraints. The Journal of Machine Learning Research, 2020.

---

### Official Review · Reviewer_iYny · 2022-07-09

**Rating:** 7
**Confidence:** 2
**Soundness:** 3 good
**Presentation:** 3 good
**Contribution:** 3 good

**Summary:**

This work studies the online convex optimization problem with 'hard' constraints. Here hard constraints mean that the violation from different rounds can not compensate for each other. The authors propose a RECOO algorithm and apply it to both the fixed constraint setting and the adversarial constraint setting. RECOO outperforms previous algorithms by achieving either a better regret or a smaller violation. At the core of RECOO is a 'rectifying' scheme applied to constraint functions. This work also considers a dynamic regret setting and provides corresponding results of their algorithms. Experiment results back up their theoretical findings.

**Questions:**

+ I do not have any specific questions.

**Limitations:**

No, the authors suggest 'We didn’t find any major limita375 tions in applying our results to constrained online convex programming.'

**Strengths And Weaknesses:**

Strengths:
+ The presentation is clear.
+ The theoretical results are new and important.
+ The experiments provide a comprehensive view of their algorithms.

Weaknesses:
+ More baseline algorithms may be mentioned in the summary section. For instance, for the fixed constraints setting, a normal online SGD method with a projection w.r.t. the constraint $g$ is also a baseline method.

---

> ### Author Response · Authors · 2022-08-02
> **Response to Reviewer iYny**
>
> We would like to sincerely thank the reviewer for the very encouraging comments. We have included more baseline algorithms such as the projected online gradient descent algorithm and online Frank-Wolfe algorithm and emphasized the advantages of RECOO over them in our revision.

---

### Official Review · Reviewer_DX8X · 2022-07-11

**Rating:** 6
**Confidence:** 3
**Soundness:** 2 fair
**Presentation:** 3 good
**Contribution:** 1 poor

**Summary:**

The problem focus on an online convex optimization problem in which the learner wants to minimize the regret and at the same time does not want to violate some (known or unknown) constraints too often. In particular, the work focus on hard constraints, i.e., such that a violation at one round cannot be compensated by a strictly feasible decision at a different round. They consider a setting with fixed known constraints, providing a regret bound of $O(\sqrt{T})$ and a violation of $O(1)$. For the setting in which the constraints are adversarial, they provide a regret bound of $O(\sqrt{T})$ and a constraint violation of $O({T}^{3/4})$. Moreover, they consider the case with strictly convex functions, proving better bounds.

**Questions:**

Let me know if I miss something about the relation with classical online convex optimization problems.

**Limitations:**

Yes

**Strengths And Weaknesses:**

The paper studies a well known important problem.
The paper is well written and provides a detailed comparison with previous works.
The paper provides new techniques that are useful to obtain better regret and violation bounds to the problem.

I have a huge concern about the significance of the result, at least for the setting with fixed constraints.
In particular, the problem can be trivially solved considering the convex decision space in which the constraint is satisfied and using a standard regret minimizer on this set. However, this approach usually requires to project on the feasible action space.
The main reason to consider a setting with relaxed constraints (soft or hard) is that in this setting is it possible to design very efficient algorithms that do not require projection on complex set (see for instance the abstract of [15]).

However, your algorithm does not seem efficient. The step Rectified decision in the Algorithm RECOO requires to compute a minimization over a function that includes a generic convex function $\hat g^+_{t-1}(\cdot)$.
Moreover, it seems that your approach is very related to standard online convex optimization problems in which the constraint g cannot be violated. In particular, if I look at your result, the constant $\epsilon$ appears only in the denominator of the regret and violation bounds. So the optimal choice for $\epsilon$ should be $\infty$. In this case $\gamma_t$ is very large and $\hat g^+_t(x)\rightarrow \infty$ when $g^+(x)$ is positive. This is a well known equivalent way to represent classical online convex optimization problems.

Post-rebuttal update: I'm partially convinced by the answers given by the authors. I agree that the algorithm proposed by the authors has a lower complexity w.r.t. standard projection based algorithms. I think that this is one of the main motivation of the work and should be highlighted. I'm still convinced that if we ignore the computational complexity motivation the problem can be solved using standard online convex optimization methods. Indeed, your method reduces to classical online convex optimization when $\epsilon$ goes to $\infty$. I think you should highlight the role of $\epsilon$ and trade of between performances and regret. Finally, I agree with the authors that the analysis is non-trivial for adversarial constraints.

---

> ### Author Response · Authors · 2022-08-02
> **Response to Reviewer DX8X**
>
> The reviewer's main concern is whether the proposed algorithm has any advantage against the projection-based methods. We'd like to address your concern from the following aspects.
>
> **Projection-based methods do not work for unknown constraints:** Projection-based methods need to know the constraint functions before decisions are made so that a safe action can be selected from the feasible set. However, in the setting we consider, the constraint function of round $t$ is revealed to the learner only after action $x_t$ is taken, so it is impossible to project an action to the unknown feasible set. Surprisingly, sublinear regret and constraint violation can still be established by using ''old'' constraint functions, and can be achieved using our unified algorithm.
>
> We do want to apologize for the confusing terminology of "fixed (known) constraints" in the current version. The setting should be fixed but *unknown* constraints. Since RECOO is designed for unifying constraints and does not need to know whether the constraints are fixed or adversarial, the two settings are just for analysis purposes where we show that for fixed constraints, RECOO improves the state-of-the-art result without explicitly utilizing the fact that the constraints are fixed. Again, without knowing that the constraints are fixed apriori, it is hard to construct a feasible set apriori. We have removed the word ''known'' to avoid confusion in our revision and added more explanation on the ''fixed'' constraints.
>
> **Our algorithm is more efficient than the projection-based method:**  As the reviewer pointed out, the projection operation in general has very high computational complexity, which is the main reason that motivates the projection-free methods, e.g., online Frank-Wolfe algorithm (see Chapter 7 in [9] or [R1]) and our related work in [26] and [30]. Note online Frank-Wolfe algorithm approximates the projection operator at each round by a linear programming with the exact same constraint functions.
>
> Our algorithm is more efficient than the projection-based algorithm and has the same complexity with [26] and [30] (note [30] is a low complexity algorithm as the title suggested). In our algorithm, we only need to solve an "almost" unconstrained optimization problem ($\mathcal X$ is usually a simple set like the box constraints). Therefore, the gradient-based methods are sufficient to find the minimizer or we might even find its close-form with the inverse operation of the function by taking ''gradient = zero''. In other words, our algorithm v.s. projection-based method is similar to unconstrained optimization problem v.s. constrained optimization problem.
>
> Besides, our algorithm is closely related to the proximal optimization method, which solves $\min_{x} (f(x) + g(x))$ with the proximal gradient method $x_{t} = Prox_{g} (x_{t-1} - \nabla f(x_{t-1})/\alpha)$ where $Prox_{g}$ is the proximal mapping of function $g$. The update of $x_t$ is equivalent to $x_{t} = argmin_x(g(x) + f(x_{t-1}) +\langle \nabla f(x_{t-1}), x-x_{t-1}\rangle + \frac{\alpha}{2} ||x - x_{t-1}||^2)$. In the online counterpart, we aim to minimize $f_t(x) + g_t(x)$ at time $t$. Since $f_t(x)$ and $g_t(x)$ are unknown when making decision at time $t$. Our algorithm approximates them with ''old'' functions $f_{t-1}(x) + \hat g^{+}_{t-1}(x)$. Interestingly, we show this design can achieve both sublinear regret and constraint violation for both fixed constraints and adversarial constraints. Since the proximal operator is applied to the old constraints and the constraints may be adversarial, the analysis is highly nontrivial. We have clarified this connection in the revision.
>
> **Discussion on [15]:** We thank the reviewer for pointing out the reference [15]. [15] is for soft constraints where the primal-dual optimization leads to efficient algorithms. As far as we know, for hard constraints, our algorithms have similar computational complexity as the existing algorithms (e.g., [30]) and more efficient than the projection-based methods.
>
> We hope the response above addresses the reviewer's concern and we sincerely hope the reviewer will reevaluate the paper based on our response.

---

> > ### Author Response · Authors · 2022-08-02
> > **Response to Reviewer DX8X**
> >
> > --------------------------- reference in our previous version --------------------------------
> >
> > [9] Elad Hazan. Introduction to online convex optimization. Foundations and Trends® in Optimization, 2016.
> >
> > [15] Mehrdad Mahdavi, Rong Jin, and Tianbao Yang. Trading regret for efficiency: online convex optimization with long term constraints. The Journal of Machine Learning Research, 2012.
> >
> > [26] Xinlei Yi, Xiuxian Li, Tao Yang, Lihua Xie, Tianyou Chai, and Karl Johansson. Regret and cumulative constraint violation analysis for online convex optimization with long term constraints. In International Conference on Machine Learning, 2021.
> >
> > [30] Hao Yu and Michael J. Neely. A low complexity algorithm with $O(\sqrt{T})$ regret and $O(1)$ constraint violations for online convex optimization with long term constraints. The Journal of Machine Learning Research, 2020.
> >
> > ------------------------------- new reference ------------------------------------------------
> >
> > [R1] Elad Hazan and Satyen Kale. Projection-free online learning. In Proceedings of International Conference on Machine Learning, 2012.

---

> ### Author Response · Authors · 2022-08-06
> **We sincerely thank for your great comments and are looking forward to your feedback.**
>
> Dear Reviewer DX8X:
>
> Since it has been a few days that the author-reviewer discussion period started, we were wondering whether you have any additional questions/comments about our response to your review comments. We greatly appreciate your time and comments and will do our best to address your concerns.

---

### Author Response · Authors · 2022-08-08
**Looking forward to your feedback.**

Dear Reviewers,

We appreciate your detailed comments again. If you have any further questions, please let us know so we can address them before the rebuttal phase ends. Thank you very much for your time!

---

### Meta-Review · Area_Chair_Yjpa · 2022-08-26

**Recommendation:** Accept
**Confidence:** Less certain

**Metareview:**

This paper provides an algorithm for online convex optimization with varying unknown constraints. Reviewers agree that the methods involved appear novel and interesting. However, the authors are strongly encouraged to add a discussion of the computational complexity of the method, which may provide the missing tradeoff for the currently free $\epsilon$ parameter.

**Award:**

No

---

### Decision · Program_Chairs · 2022-09-14

Accept